# Assessment of Mixed *Plasmodium falciparum* *sera5* Infection in Endemic Burkitt Lymphoma: A Case-Control Study in Malawi

**DOI:** 10.3390/cancers13071692

**Published:** 2021-04-02

**Authors:** Nobuko Arisue, George Chagaluka, Nirianne Marie Q. Palacpac, W. Thomas Johnston, Nora Mutalima, Sally Peprah, Kishor Bhatia, Eric Borgstein, George N. Liomba, Steve Kamiza, Nyengo Mkandawire, Collins Mitambo, James J. Goedert, Elizabeth M. Molyneux, Robert Newton, Toshihiro Horii, Sam M. Mbulaiteye

**Affiliations:** 1Research Center for Infectious Disease Control, Research Institute for Microbial Diseases, Osaka University, Osaka 565-0871, Japan; arisue@biken.osaka-u.ac.jp; 2Departments of Pediatrics and Surgery, College of Medicine, University of Malawi, Private Bag 360, Chichiri, Blantyre 3, Malawi; gchagaluka@medcol.mw (G.C.); eborgstein@medcol.mw (E.B.); profngliomba@yahoo.com (G.N.L.); skamiza@medcol.mw (S.K.); nmkandawire@medcol.mw (N.M.); emmolyneux@gmail.com (E.M.M.); 3Department of Malaria Vaccine Development, Research Institute for Microbial Diseases, Osaka University, Osaka 565-0871, Japan; nirian@biken.osaka-u.ac.jp (N.M.Q.P.); horii@biken.osaka-u.ac.jp (T.H.); 4Epidemiology and Cancer Statistics Group, Department of Health Sciences, University of York, York YO10 5DD, UK; tom.johnston@york.ac.uk (W.T.J.); nora.mutalima@monashhealth.org (N.M.); Robert.Newton@york.ac.uk (R.N.); 5Cancer Epidemiology Unit, University of Oxford, Oxford OX3 7LF, UK; 6Infections and Immunoepidemiology Branch, Division of Cancer Epidemiology and Genetics, National Cancer Institute, National Institutes of Health, Bethesda, MD 20892, USA; sally.peprah@nih.gov (S.P.); dnadoc@hotmail.com (K.B.); jamesjgoedert@gmail.com (J.J.G.); 7National Health Sciences Research Committee, Research Department, Ministry of Health, P.O. Box 30377, Capital City, Lilongwe 3, Malawi; collins.mitambo@health.gov.mw

**Keywords:** *Plasmodium falciparum*, Burkitt lymphoma, Epstein Barr virus, epidemiology, Africa, complexity of infection

## Abstract

**Simple Summary:**

*Plasmodium falciparum**(Pf)* infection is a risk factor for endemic Burkitt lymphoma (eBL), the commonest childhood cancer in Africa, but the biomarkers of *Pf* infection that predict this risk are unknown. There is some evidence that the genetic complexity of *Pf* infection may be a risk factor. In 200 children with versus 140 without eBL in Malawi, this study compared variants of the malaria parasite, focusing on *Pfsera5*, a gene that codes for malaria protein that an infected person’s antibodies target to suppress the parasite. Multiple *Pfsera5* variants, which arise when the parasite is not suppressed, were found in 41.7% of eBL children versus 24.3% of other local children, meaning that eBL risk was increased 2.4-fold with multiple *Pfsera5* variants. No specific type of variant was related to eBL risk. Research to quantify malaria parasite variants and to clarify the host immune response needed to control variant infections may yield a test to predict eBL risk.

**Abstract:**

Background: Endemic Burkitt lymphoma (eBL) is the most common childhood cancer in Africa and is linked to *Plasmodium falciparum* (*Pf*) malaria infection, one of the most common and deadly childhood infections in Africa; however, the role of *Pf* genetic diversity is unclear. A potential role of *Pf* genetic diversity in eBL has been suggested by a correlation of age-specific patterns of eBL with the complexity of *Pf* infection in Ghana, Uganda, and Tanzania, as well as a finding of significantly higher *Pf* genetic diversity, based on a sensitive molecular barcode assay, in eBL cases than matched controls in Malawi. We examined this hypothesis by measuring diversity in *Pf-*serine repeat antigen-5 (*Pfsera5*), an antigenic target of blood-stage immunity to malaria, among 200 eBL cases and 140 controls, all *Pf* polymerase chain reaction (PCR)-positive, in Malawi. Methods: We performed *Pfsera5* PCR and sequencing (~3.3 kb over exons II–IV) to determine single or mixed *Pf*SERA5 infection status. The patterns of *Pfsera5* PCR positivity, mixed infection, sequence variants, and haplotypes among eBL cases, controls, and combined/pooled were analyzed using frequency tables. The association of mixed *Pfsera5* infection with eBL was evaluated using logistic regression, controlling for age, sex, and previously measured *Pf* genetic diversity. Results: *Pfsera5* PCR was positive in 108 eBL cases and 70 controls. Mixed *Pf*SERA5 infection was detected in 41.7% of eBL cases versus 24.3% of controls; the odds ratio (OR) was 2.18, and the 95% confidence interval (CI) was 1.12–4.26, which remained significant in adjusted results (adjusted odds ratio [aOR] of 2.40, 95% CI of 1.11–5.17). A total of 29 nucleotide variations and 96 haplotypes were identified, but these were unrelated to eBL. Conclusions: Our results increase the evidence supporting the hypothesis that infection with mixed *Pf* infection is increased with eBL and suggest that measuring *Pf* genetic diversity may provide new insights into the role of *Pf* infection in eBL.

## 1. Introduction

Endemic Burkitt lymphoma (eBL) is an aggressive B-cell non-Hodgkin lymphoma (NHL), first described in African children by Denis Burkitt in 1958 [1]. High incidence of eBL overlaps with holoendemic *Plasmodium falciparum* (*Pf*) infection in Africa [2], and it is one of the commonest childhood cancers [3] in countries with high malaria endemicity, including Malawi, Uganda, Tanzania, and Kenya [4]. The eBL incidence is 30-fold higher in these countries than in non-malaria endemic countries [5]. The correlation of eBL with malaria in population-based and individual-level studies, the findings of altered anti-*Pf* antibody levels in eBL cases compared to healthy children [6,7,8], and the reduced frequency of genetic variants that protect against malaria in eBL cases compared to healthy children [9,10] suggest that malaria infection may be related to eBL risk. Biologically, *Pf* may increase eBL risk directly by stimulating the polyclonal expansion of B lymphocytes, triggering chromosomal instability in B cells [11,12], or indirectly, by impairing immunologic control of the Epstein–Barr virus (EBV), a known carcinogen for eBL [13].

The biomarkers of *Pf* related to eBL are unknown. A correlation study of 2602 cases in Ghana, Uganda, and Tanzania with published biomarkers (parasitemia, parasite density, and genetic diversity of parasites) of *Pf* infection in the same countries [14] was the first to suggest that *Pf* genetic diversity may be related to eBL, based on a significant association between age-specific patterns of eBL and *Pf* genetic diversity. This hypothesis was evaluated among 303 eBL cases and 274 controls in Malawi by measuring *Pf* diversity using a *Pf*3D7 molecular barcode assay [15]. The results showed a significantly higher mean *Pf* genetic diversity score in the eBL cases than the controls [16]. Although these two studies represent the only evidence linking *Pf* genetic diversity with eBL, the results may be valid because they are based both on population- and individual-level designs, and an assessment of *Pf* diversity using a sensitive and specific molecular assay.

In the present study, we investigated whether the presence of mixed infection at the *Pf* serine repeat antigen-5 (*Pfsera5*) locus (on chromosome 2) [17] is associated with eBL in Malawi. *Pfsera5* has been identified as a blood-stage vaccine candidate [18,19], in part because antibodies targeting *Pfsera5*, measured using SE36—a recombinant molecule of *Pf*SERA5 [20]—were protective against malaria and eBL [7]. The potential as a vaccine target also prompted interest in the genetic diversity at this locus. *Pfsera5* codes for a 120-kDa precursor (Figure 1A), which is critical to *Pf* blood-stage infection (and egress from the parasitophorous vacuole) and is an antigenic target of blood-stage immunity to malaria [21]. The *Pf*SERA5 precursor is processed by removing the signal peptide and the protein trimmed into three fragments: the P47, P56, and P18 kDa domains; P56 is further processed to P50 and P6 fragments [22,23,24,25]. *Pfsera5* sequence diversity is introduced by insertions and deletions in the P47 fragment in the protein domain containing the octamer and serine repeats (where protective epitopes are located) [26], and by point mutations in a stretch without repeats [27], but the association of *Pfsera5* diversity with eBL is unknown.

## 2. Materials and Methods

### 2.1. Study Patients

The study was conducted in the Infections and Childhood Cancer study in Malawi [28]. Briefly, children aged 0 to 15 years diagnosed with cancers (eBL, leukemia, Hodgkin lymphoma, neuroblastoma, rhabdomyosarcoma, Ewing’s sarcoma, primitive neuroectodermal tumor, and Wilms’ tumor) were enrolled at the Queen Elizabeth Hospital in Blantyre, Malawi between July 2005 and August 2010. All children were reviewed by one investigator (EM) to verify clinical diagnosis. Confirmation by histology, cytology, or other laboratory investigations was done when possible. Children with eBL were coded as cases and those with another diagnosis were coded as controls [16]. Children with HIV infection and those with Kaposi sarcoma were excluded. The current study included only children who were *Pf*-infected based on polymerase chain reaction (PCR) performed on samples taken at the time of diagnosis, before initiating cancer treatment targeting a 519-base pair (bp) segment of the PF07-0076 locus [16].

### 2.2. Ethics Review

Ethical approval was given by the Malawian College of Medicine Research and Ethics Committee (P.03/04/277R) and the Office of Human Subjects Research at the National Institutes of Health (Exempt #: 4742). The parents/guardians of the children gave written informed consent.

### 2.3. Laboratory Methods

#### 2.3.1. *Pfsera5* Gene Amplification

The *Pf*SERA5 amplification was performed on residual genomic DNA extracted from coded/masked whole blood samples taken at the time of diagnosis before initiating cancer treatment, using a QIAamp Blood DNA Kit (Qiagen, Inc., Valencia, CA, USA) at Osaka University [16]. The *Pf*SERA5 PCR amplification targeted the ~3.3 kb region in exons II–IV using the specific primers sera5-5F0 and sera5-3R0 (Appendix A). The sequence before and after exon I (encoding the signal peptide) is AT-rich and contains poly-A residues, which makes it difficult to successfully sequence; thus, this region was excluded in our analysis. PCR amplification was carried out in a 25-μL reaction mixture containing 0.4 μM each of forward and reverse primers, 0.4 mM of deoxyribonucleotides (dNTP), 0.5 units of KOD FX Neo polymerase (TOYOBO Co., Ltd., Osaka, Japan), 12.5 μL of 2× PCR buffer, and 1 μL of genomic DNA solution. The PCR conditions were as follows: 95 °C for 2 min, 40 cycles of 95 °C for 30 s, 59 °C for 30 s, and 68 °C for 4 min. A 2-μL aliquot of the PCR product was used as a template for a second PCR amplification in a 25-μL reaction mixture using the primers sera5-5F3 and sera5-3R2 (Appendix A) under the same thermocycler conditions. Samples that were not successfully amplified using this primer set were re-tested using nested PCR primers targeting half the original length of the two fragments (a 5′ half and a 3′ half). The nested primer set used for the 5′ half fragments were sera5-5F0 and sera5-R0 for the first PCR, and sera5-5F3 and sera5-R0 for the nested PCR; and for the 3′ half fragments, sera5-F1 and sera5-3R0 were used for the first PCR, followed by sera5-F2 and sera5-3R2 (Appendix A). The nested PCR conditions were the same as for the full-length PCR, but the extension time was shortened to 2 min. There were 12 duplicate samples included for quality control. The results were concordant for PCR and *PfSERA5* single or mixed infection in nine samples at the original testing, and three samples were resolved after re-testing; two were PCR-negative and one was a single infection.

#### 2.3.2. *Pfsera5* Nucleotide Sequencing

The PCR products were purified using a QIAquick PCR Purification kit (QIAGEN, Hilden, Germany) according to the manufacturer’s instructions. Purified DNA fragments were eluted in 30 μL of Tris-EDTA buffer (TE). The optical density was measured with a NanoDrop (Thermo Fisher Scientific, Wilmington, DE, USA) and the DNA concentration was adjusted to 0.026 μg/μL for the 3.3 kb fragments and 0.013 μg/μL for the 5′-half and 3′-half fragments using TE. At this concentration, 1 μL was suitable for performing one sequencing reaction. *Pfsera5* DNA sequencing was performed directly using The BigDye^®^ Terminator v3.1 Cycle Sequencing Kit and 3130xI Genetic Analyzer (Applied Biosystems, Foster City, CA, USA). Sequencing primers were designed to cover target regions in both directions (Appendix A). The nucleotide sequences generated during the current study are available through public database such as DDBJ and NCBI (Accession numbers: LC606291–LC606405). The sequence results without overlapping peaks on the electropherograms were interpreted as being single parasite infections; otherwise, the sample was recorded as having a mixed parasite infection. The *Pf*SERA5 sequence variations in single parasite infections were analyzed to get insights into variation in populations.

#### 2.3.3. Sequence Alignment and *Pf* Population Genetics Analyses

The *Pf*SERA5 nucleotide sequence obtained from each sample was aligned to the *PfSERA5* 3D7 strain using CLUSTALW implemented in GENETYX^®^ ver. 15 (GENETYX Co., Ltd., Tokyo, Japan) and the alignment was manually inspected to ensure alignment accuracy. Analyses were done for the entire sequence and for specific protein domains: the 2562 bp nucleotide sequence includes a stretch without repeats (a non-repeat region, or NonR); the stretch of octamer repeats (OctR) and the stretch with serine repeats (SerR) correspond to amino acid positions 87–193 and 251–997, respectively [27]. The sequence information from the NonR was translated into its 854 amino acid (aa) sequences to identify nonsynonymous (variations that result in changes in an aa sequence) versus synonymous (no change in an aa sequence). To gain insights into *Pfsera5* variations in samples from nearby Tanzania or countries with a high eBL incidence (e.g., Ghana), we accessed the *Pfsera5* sequence from 55 asymptomatic donors in the Rufiji River Delta in Tanzania sampled in 1993, 1998, and 2003 (Accession numbers: AB634928–AB634982) [29], and from 33 children in three villages in Ghana sampled in 2004 (Accession numbers: AB634983–AB635015) [29].

Haplotype diversity (Hd) and nucleotide diversity (π, the average number of nucleotide differences per site between two sequences in all possible pairs in the sample population) were calculated using DnaSP v5.10.01 [30]. The difference between the numbers of synonymous substitutions per synonymous site (dS) and of nonsynonymous substitutions per nonsynonymous site (dN) was calculated by the Nei and Gojobori method [31], implemented in MEGA X, with the Jukes and Cantor correction. The statistical significance of the difference between dN and dS was estimated using the MEGA Z-test [32]. Higher dS may suggest that active, synonymous amino acid substitutions that are neutral or can boost parasite fitness are more likely to be retained or accumulated, while non-synonymous amino acid substitutions that reduce parasite fitness are removed or reduced in frequency. Wright’s fixation index *Fst* was calculated to assess the genetic differentiation of *Pfsera5* among the parasite circulating in the study population [33]. The pairwise *Fst* between parasite populations was calculated using Arlequin v3.5 [34]. A small *Fst* indicates that the parasite allele frequencies within the study populations are similar, whereas a large *Fst* indicates that the allele frequencies are different, and the study populations do not share genetic diversity.

#### 2.3.4. Statistical Analyses

Patterns of *Pfsera5* PCR positivity, mixed *Pfsera5* infection, and *Pfsera5* haplotypes and nucleotide diversity in eBL cases or controls, separately or combined/pooled, were analyzed for the entire *Pfsera5* sequence and by domain (OctR, SerR, and NonR). The results were categorized into known haplotypes; the many “rare haplotypes” (defined among Malawi controls as SerR haplotypes observed in <10% or OctR haplotypes observed in <20% of the samples) were grouped and then compared to the common haplotypes. This grouping is post hoc, but it is useful for exploring pairwise comparisons of common versus rare haplotypes in eBL cases and controls, as well as haplotype patterns observed in Ghana and Tanzania. Differences across sample groups were evaluated using Fisher’s exact test. Because age is an important predictor of exposure and immunity to malaria, the data were explored by age (<6 versus ≥6 years of age).

The association of mixed *PfSERA5* infection with eBL was assessed by calculating odds ratios and 95% confidence intervals (OR 95% CI) using logistic regression. Bivariate ORs (bORs) were obtained by adjusting separately for age, sex, and the *Pf* genetic diversity score as confounders [16], before including all these variables into one model for multivariate adjustment (aOR), but the results were not adjusted for multiple comparisons; thus, two-sided *p*-values < 0.05 were considered statistically significant. However, other comparisons must be interpreted cautiously for hypothesis generation.

## 3. Results

### 3.1. Characteristics of Study Subjects

The current study evaluated 341 *Pf* PCR-positive participants. One participant was >17 years and excluded from subsequent analyses. The remaining participants included 200 eBL cases and 140 controls (Table 1). Most participants were from Malawi (*n* = 327), but 13 (10 eBL cases and 3 controls) were from Mozambique, a neighboring country. Because of the small sample size of the children originating in Mozambique, the sequence data from those children were analyzed together with those from Malawi, where all children were enrolled. The three leading diagnoses among the controls were renal tumors (*n* = 28), non-eBL lymphoma and leukemia (*n* = 35, including 9 with leukemia), and soft tissue tumors (*n* = 19). The eBL cases were slightly older than the controls, but the difference was not statistically significant (7.2 years versus 6.9 years, *p* = 0.44). However, the eBL cases were more likely to be aged 6–10 years than the controls (50.5% versus 25.7%, heterogeneity <0.001). The male-to-female proportion of eBL cases and controls was not statistically different in this set (62.5% versus 58.6% in females, *p* = 0.50).

*Pfsera5* PCR positivity was detected in 108 (54.0%) eBL cases and 70 (50.0%) controls (Table 1). The PCR product was insufficient in one control sample and cannot be conclusively ascertained if the sample is a single or mixed *Pfsera5* infection. Likewise, not all *Pf* positive samples were suitable for amplification of the near full-length *Pfsera5*, due mainly to the quality of samples available for the study.

### 3.2. Association of Mixed Pfsera5 Infection with eBL

*Pfsera5* sequences revealed mixed infection in 45 (41.7%) eBL cases and 17 (24.3%) controls. Mixed *Pfsera5* infection was associated with eBL risk in crude analysis (cOR of 2.18, 95% CI of 1.12–4.26: Table 1; Appendix A) and in bivariate analyses adjusting for gender (bOR of 2.18, 95% CI of 1.12–4.26), *Pf* diversity score (bOR of 2.23, 95% CI of 1.10–4.53), *Pf* DNA copy number (bOR of 1.95, 95% CI of 0.96–3.95), and age group (<6, 6–10, 11+ years) (bOR of 2.44, 95% CI of 1.12–4.26). The association of mixed *Pfsera5* infection with eBL remained in models adjusting for all the variables above (aOR of 2.40, 95% CI of 1.11–5.17).

### 3.3. Pfsera5 Haplotype Diversity in eBL Cases and Controls in Malawi

Analysis of Hd and π were calculated for 115 samples in Malawi and 88 individuals in Tanzania and Ghana with single *Pfsera5* infection. As shown in Figure 1B, the parasites harbored variants at 29 nucleotide positions in the sequenced 2562 bp NonR region; 17 led to nonsynonymous amino acid changes (highlighted orange) and 12 led to synonymous amino acid changes (highlighted green). The numbers of haplotypes are shown in Table 2. Analysis of the entire *Pfsera5* sequence yielded 96 haplotypes (88 haplotypes at the amino acid sequence level), 34 in the OctR, 42 in the SerR, and 26 in the NonR regions (Table 2, detailed in Appendix A). The Hd for the entire *Pfsera5* sequence was 0.993 in eBL cases, 0.996 in the controls, and 0.991 and 0.996 in the samples from Tanzania and Ghana, respectively (Table 2). When Hd was considered by *Pfsera5* regions, it was lowest for the NonR region and moderately low for the OctR region; Hd was similar for the entire *Pfsera5* and the SerR region, which bears the protective epitopes of *Pf*SERA5 [26]. Hd in the NonR region was low in Tanzania (0.362) and Ghana (0.472) compared to that in the eBL cases and controls combined (0.545) (Malawi and Mozambique).

### 3.4. Patterns of Common Versus Rare Pfsera5 Haplotypes in eBL Cases and Controls

In analyses restricted to those with a single *Pfsera5* infection, the frequency of the common haplotypes in the OctR and SerR regions was similar in children with eBL and controls. The common haplotypes in the OctR and SerR regions were more frequent in the combined eBL cases and the controls in Malawi and Mozambique than in Ghana and Tanzania (Figure 2, Appendix A, *p* < 0.05).

### 3.5. Genetic Differentiation between Parasite Populations

Table 3 shows nucleotide diversity (π) in the 2562 bp NonR and Table 4 shows the location of the 29 polymorphic sites (12 synonymous and 17 non-synonymous). There are 23 new variations, that is, variations not previously reported, while 6 (5 non-synonymous and 1 synonymous) have previously been reported [21], but the specific variations were not different between the cases and controls. The eBL cases, controls, and individuals in Tanzania and Ghana had similar nucleotide diversity (π). However, the current eBL cases and controls (combined, Malawi and Mozambique) had dS > dN while the opposite pattern (i.e., dN > dS) was observed in the samples from Tanzania (*p* < 0.06) and Ghana (*p* = 0.0034) (Table 3). *Fst* analysis suggested genetic differentiation (*p* < 0.05) in the OctR and SerR, but not the NonR regions for parasites in the eBL cases and controls (combined) and in individuals in Ghana (Table 5). However, *Fst* analysis did not support evidence of genetic differentiation between parasites in Malawi and Mozambique versus those in Tanzania and Ghana.

## 4. Discussion

We investigated the hypothesis that *Pf* genetic diversity, based on the detection of mixed *Pfsera5* infection, is associated with eBL in Malawi [16]. Our findings of 2.40-fold higher odds of mixed *Pf*SERA5 infection in eBL cases than controls, which remained statistically significant after the adjustment of confounders, including gender, age group, *Pf* DNA copy, and *Pf* genetic diversity [16], led us to reject the null hypothesis. These results strengthen the evidence that *Pf* genetic diversity may be related to eBL, as highlighted in our earlier ecological study in Uganda, Tanzania, and Ghana [14] and in our case-control in Malawi using a sensitive molecular assay [16].

There is accumulating evidence that mixed *Pf* infection may be a molecular surrogate of chronic asymptomatic *Pf* infection in high malaria transmission areas [35]. The consequences for malaria risk appear similar to earlier studies, that is, the risk for malaria (infection and symptoms) is high in children below age 5 [36,37] but low in children above age 5 [38], consistent with the level of acquired immunity against malaria at those ages [39]. Thus, in older children, mixed infection appears to be a molecular marker of asymptomatic infection [35], incomplete clearance of parasites [40], clinical and parasite immunity [41], and, according to our results, eBL risk. A recent finding of a median of the multiplicity of infection of six haplotypes in infective mosquitoes suggests that mixed infections may arise from co-transmission of genetically diverse oocysts from a single mosquito bite [35]. Super-infection, that is, the inoculation of different parasites from different mosquitoes, is also possible [42] in partially immune people living in malarious, non-sheltered environments [43].

Chronic, asymptomatic, low-density parasite infection has been reported in eBL cases and may precede the disease [40]. Asymptomatic infection is due, in part, to antibodies that target parasite proteins, particularly those that are secreted and embedded in infected red blood cell (iRBC) membranes, thereby blocking sequestration and promoting splenic clearance of iRBCs [44,45,46], and thus suppressing parasite density and clinical symptoms. However, because asymptomatic infections are sub-clinical, they remain untreated for prolonged periods; the associated chronic immune response may lead to eBL as a rare complication in children unsheltered from malaria [8]. While a strong immune response may explain the reduced frequency of symptomatic malaria and lower parasite densities in eBL cases than other children in the same region [40], we speculate that *Pfsera5* may contribute to the molecular camouflage that facilitates a tolerance of low-grade infection and sets the stage for progression to cancer. *Pf*SERA5 N-terminal domain decorates the surface of the merozoite where it tightly binds to a host serum protein, vitronectin, which in turn binds to another host protein that camouflages the parasite from the host immune system [47], potentially reducing pressure for mutations in the *Pfsera5* gene. However, this immune response is not completely effective, resulting in immune tolerance, high antibody titers against *Pf*SERA5, and low-grade parasitemia [20]. There is still a dearth of understanding in the chronic nature of malaria infection and how it amplifies or down-regulates the immune response, although the continuous insult by low-grade, often genetically complex variants (mean of 4) [35] likely increases the risk of genetic instability in B cells and their progression to eBL [11]. In agreement with this hypothesis, the spleen is frequently enlarged in children with asymptomatic *Pf* infection [48], whom we hypothesize are the population at risk of eBL [40]. Plausibly, a robust B-cell response promotes survival against malaria [49], on one hand by suppressing *Pf* positivity in eBL [40,43,50], while promoting eBL risk in children with asymptomatic infection, on the other hand.

The use of *Pfsera5* molecular data expands our knowledge of variants in *Pf* field isolates in countries where eBL incidence is high [4]. Consistent with previous reports [27], we observed that *Pfsera5* is highly conserved with limited evidence of genetic differentiation, consistent with the frequent gene flow in Africa. We also observed that *Pfsera5* diversity was not unique to Malawi and Mozambique or to eBL cases. These results are reassuring because our comparisons are based on samples obtained using varied, potentially biased methods. The samples from Ghana and Tanzania are from specific communities living in relatively small geographical areas, whereas the Malawi and Mozambique samples are from a much larger geographical area. Thus, the observations that no specific *Pf* variants or haplotypes are associated with eBL and that both eBL cases and controls in Malawi harbor common haplotypes are consistent with both groups being exposed to similarly high environmental pressure of *Pf* infection [35]. However, we note that rare haplotypes may be harder to detect because they exist at lower submicroscopic levels. Our findings of a lower frequency of the common haplotypes in Tanzania and Ghana than in Malawi and Mozambique may be due to geographic confounding, as noted above.

Our research suggests that molecular markers of *Pf* genetic diversity may complement the search for biomarkers of *Pf* infection related to eBL risk. The general finding that *Pf*SERA5 does not show antigenic variation [20] suggests that *Pfsera5* might be a good tool to characterize the genetic complexity of parasite populations in eBL cases and controls. However, this should be balanced against the likelihood that the sample size will shrink because amplifying about 3.2 Kb of *Pfsera5* requires higher quality gDNA than amplifying the 519 bp that we used to confirm *Pf* infection in all our samples. This explains why our sample size shrunk by 50%. Other limitations of our study include the use of cross-sectional samples from a case-control study, and the testing of one sample from a single time point, which underestimates parasite diversity [51]. These limitations are balanced by our hypothesis-driven approach, our focus on diversity at a locus that is an antigenic target of blood-stage immunity to malaria, and the use of cancer controls, in whom referral bias and reverse-causality biases are minimized. Our study included only children who were confirmed to have *Pf* infection by PCR [16], which increases the validity of our assumption that all children were exposed to infection.

## 5. Conclusions

Our study provides new evidence for the association of mixed *Pf* infection with eBL risk in Malawi. Further research utilizing molecular markers of *Pf* infection may lead to the discovery of biomarkers of eBL, severe malaria, and/or asymptomatic infection.

## Figures and Tables

**Figure 1 cancers-13-01692-f001:**
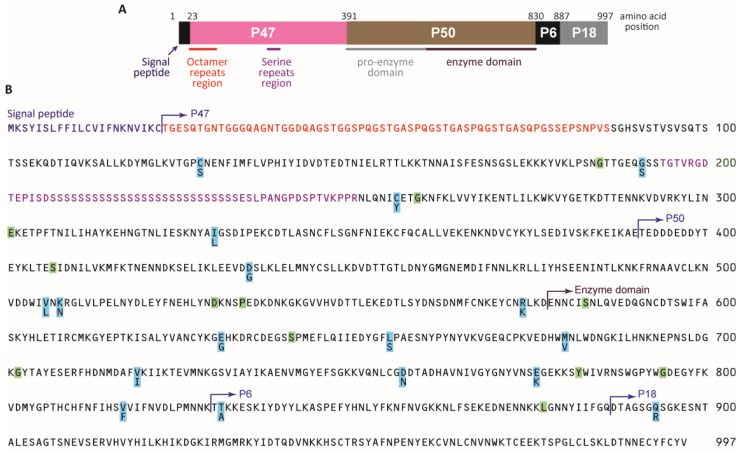
Structure, divergence, and polymorphism of the *P. falciparum sera5* gene. (**A**) Schematic structure SERA5. Numbers are the amino acid positions of the SERA5 gene in the *P. falciparum* 3D7 strain. (**B**) Amino acid sequence of SERA5 from *P. falciparum* 3D7. Polymorphic amino acid residues are highlighted in light blue. Positions with synonymous amino acid changes are highlighted in light green. The signal peptide region is shown in purple. Octamer repeats are shown in red. The serine repeat region with a 13-mer insertion/deletion region and a dimorphic 17-mer region are shown in pink.

**Figure 2 cancers-13-01692-f002:**
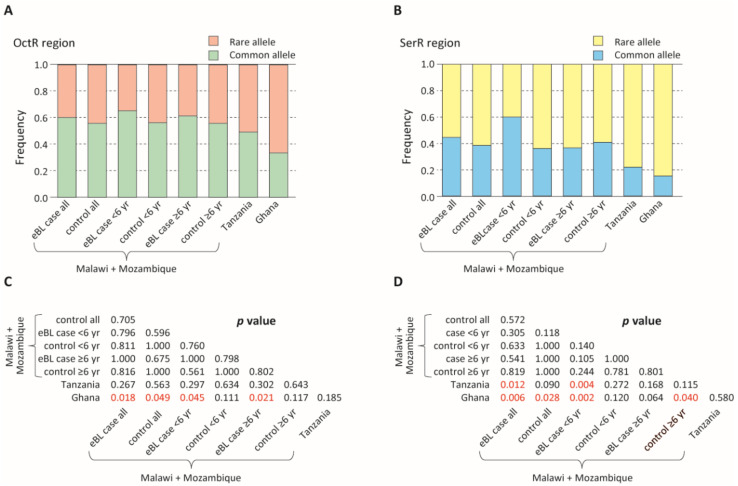
Comparison of the proportion of rare and common alleles among endemic Burkitt lymphoma (eBL) cases and controls. (**A**) Comparison of the proportion of the rare and common alleles in the octamer repeat (OctR) region. Alleles with a frequency >0.2 are defined as common alleles, while the rest are defined as rare alleles. (**B**) Comparison of the proportion of the rare and common alleles in the serine repeat (SerR) region. Alleles with a frequency >0.1 are defined as common alleles, while the rest are defined as rare alleles. (**C**,**D**) *p*-values of the Fisher’s exact test (C: OctR region; D: SerR region). *p* < 0.05 are shown in red.

**Table 1 cancers-13-01692-t001:** Characteristics of Malawi and Mozambique samples used in this study.

Subject Characteristics	Sample Size	Gender	Age Group					*Sera5* PCR		*Sera5* Sequence		
		Male (%)	Missing Age	Mean (SD)	0–5yrs	6–10yrs	11–15yrs ^*^	Negative	Positive	Single	Mixed	Indeterminate ^†^
**Case Status**												
Cases	200	125 (62.5)	8	7.2 (2.8)	69 (34.5)	101 (50.5)	22 (11.0)	92 (46.0)	108 (54.0)	63 (58.3)	45 (41.7)	-
Controls	140	82 (58.6)	5	6.9 (4.4)	66 (47.1)	36 (25.7)	33 (23.6)	70 (50.0)	70 (50.0)	52 (74.3)	17 (24.3)	1 (1.4)
Association with eBL												
Odds ratio [OR 95% CI]	-	-	-	-	-	-	-	-	-	Ref	2.18 (1.12, 4.26)	-
Adjusted OR ^‡^ [95% CI]	-	-	-	-	-	-	-	-	-	Ref	2.40 (1.11, 5.17)	-
**Diagnose in Controls**												
Leukemias	9	3 (33.3)	0	7.6 (4.5)	4 (44.4)	2 (22.2)	3 (33.3)	7 (77.8)	2 (22.2)	1 (50.0)	1 (50.0)	-
Lymphomas	26	18 (69.2)	1	10.2 (4.1)	4 (16.0)	8 (32.0)	13 (52.0)	13 (50.0)	13 (50.0)	11 (84.6)	2 (15.4)	-
Neuroblastomas	8	5 (62.5)	0	7.0 (3.9)	3 (37.5)	4 (50.0)	1 (12.5)	6 (75.0)	2 (25.0)	2 (100.0)	0 (0.0)	-
Retinoblastoma	9	3 (33.3)	0	4.0 (2.8)	8 (88.9)	0 (0.0)	1 (11.1)	5 (55.6)	4 (44.4)	2 (50.0)	2 (50.0)	-
Renal tumors	28	21 (75.0)	0	3.6 (2.9)	24 (85.7)	3 (10.7)	1 (3.6)	8 (28.6)	20 (71.4)	15 (75.0)	5 (25.0)	-
Hepatic tumors	6	3 (50.0)	0	7.6 (2.7)	2 (33.3)	3 (50.0)	1 (16.7)	2 (33.3)	4 (66.7)	3 (75.0)	1 (25.0)	-
Bone tumors	2	1 (50.0)	0	10.3 (5.4)	0 (0.0)	1 (50.0)	1 (50.0)	1 (50.0)	1 (50.0)	1 (100.0)	0 (0.0)	-
Soft tissue sarcomas	19	14 (73.7)	1	7.8 (4.2)	8 (44.4)	5 (27.8)	5 (27.8)	11 (57.9)	8 (42.1)	6 (75.0)	2 (25.0)	-
Germ cell tumors	8	3 (37.5)	0	6.8 (4.0)	4 (50.0)	2 (25.0)	2 (25.0)	3 (37.5)	5 (62.5)	3 (60.0)	2 (40.0)	-
Epithelial tumors	2	0 (0.0)	0	6.7 (7.8)	1 (50.0)	0 (0.0)	1 (50.0)	1 (50.0)	1 (50.0)	1 (100.0)	0 (0.0)	-
Other tumors	1	0 (0.0)	0	13.5 (-)	0 (0.0)	0 (0.0)	1 (100.0)	1 (100.0)	0 (0.0)	0 (0.0)	0 (0.0)	
Non-malignancies	10	6 (60.0)	0	7.3 (4.7)	3 (30.0)	5 (50.0)	2 (20.0)	5 (50.0)	5 (50.0)	4 (80.0)	0 (0.0)	1 (20.0)
Not well specified	12	5 (41.7)	3	6.2 (4.6)	5 (55.6)	3 (33.3)	1 (11.1)	7 (58.3)	5 (41.7)	3 (60.0)	2 (40.0)	-

* Two controls between 16 and 16.2 years; ^†^ Unclear if positive infection detected was single or mixed; ^‡^ Odds ratio adjusted for diversity score, DNA copy number, gender, and age.

**Table 2 cancers-13-01692-t002:** Haplotype diversity of the entire sequence, and the octamer repeat (OctR), serine repeat (SerR), and non-repeat (NonR) 2562 base pair (bp) regions of the *P. falciparum SERA5* gene.

Country	*n*	Entire Sequence		OctR Region		SerR Region		NonR 2562 bp Region	
		No. of Haplotypes	Hd ± SD	No. of Haplotypes	Hd ± SD	No. of Haplotypes	Hd ± SD	No. of Haplotypes	Hd ± SD
Malawi and Mozambique	115	96 (88)	0.994 ± 0.003	34 (34)	0.809 ± 0.032	42 (39)	0.935 ± 0.011	26 (16)	0.545 ± 0.011
eBL case	63	54 (50)	0.993 ± 0.005	19 (19)	0.786 ± 0.047	28 (27)	0.928 ± 0.019	16 (9)	0.551 ± 0.075
Control	52	47 (45)	0.996 ± 0.002	21 (21)	0.839 ± 0.040	26 (23)	0.945 ± 0.016	14 (9)	0.544 ± 0.083
Tanzania	55	44 (42)	0.991 ± 0.006	21(21)	0.804 ± 0.054	30 (30)	0.968 ± 0.010	11 (9)	0.362 ± 0.084
Ghana	33	31 (29)	0.996 ± 0.009	16 (16)	0.873 ± 0.048	23 (23)	0.968 ± 0.018	8 (7)	0.472 ± 0.106

The numbers of amino acid sequence variations are shown in parenthesis. eBL: endemic Burkitt Lymphoma, Hd: Haplotype diversity, SD: standard deviation

**Table 3 cancers-13-01692-t003:** Nucleotide divergence and polymorphism in the non-repeat 2562 bp sequence regions.

Country	*n*	No. of Polynmorphic Sites	No. of Substitutions		Nucleotide Diversity	Codon-Based Evolutionary Divergence		
			Synonymous	Non-Synonymous	π ± SD	dS ± SE	dN ± SE	*p* Value
Malawi and Mozambique	115	29	12	17	0.00029 ± 0.00004	0.00050 ± 0.00012	0.00024 ± 0.00008	dS > dN: 0.0595
eBL case	63	17	8	9	0.00028 ± 0.00005	0.00055 ± 0.00017	0.00021 ± 0.00009	dS > dN: 0.0450
Control	52	16	6	10	0.00031 ± 0.00006	0.00044 ± 0.00017	0.00027 ± 0.00011	dS > dN: 0.1943
Tanzania	55	11	1	10	0.00021 ± 0.00007	0.00007 ± 0.00007	0.00025 ± 0.00008	dN > dS: 0.0549
Ghana	33	10	1	10	0.00032 ± 0.00011	0.00000 ± 0.00000	0.00040 ± 0.00015	dN > dS: 0.0034

π: nucleotide diversity, SD: standard deviation, dS: the numbers of synonymous substitutions per synonymous site, SE: standard error, dN: nonsynonymous substitutions per nonsynonymous site, eBL: endemic Burkitt Lymphoma.

**Table 4 cancers-13-01692-t004:** Sequence variation in *P. falciparum sera5* in the non-repeat 2562 bp region. Nucleotide and amino acid positions are numbered after the *P. falciparum* 3D7 sequence. The number of isolates with variations are shown. Variations shared with previously reported *P. falciparum sera5* sequences in PlasmoDB (https://plasmodb.org/plasmo/ accessed on 23 October 2020) and NCBI (https://www.ncbi.nlm.nih.gov/ accessed on 23 October 2020) databases are shown in bold font.

(A) Non-synonymous change.																											
**Nucleotide Position**	3 8 3	4 7 5	5 6 2	5 6 6	**5** **7** **1**	7 6 7	7 8 2	7 9 2	**9** **8** **8**	1 3 0 4	**1** **5** **1** **6**	1 5 2 4	1 6 1 3	1 7 2 1	1 8 9 2	1 9 6 4	**2** **0** **3** **8**	2 0 6 5	2 1 5 5	2 2 6 9	2 3 2 5	**2** **3** **2** **6**	2 4 4 9	2 4 9 1	2 6 7 8	2 8 0 1	2 8 3 3
**Common**	G	K	G	A	**G**	G	G	G	**A**	A	**G**	G	A	G	A	T	**A**	T	G	G	C	**G**	G	A	A	G	A
**Variation**	C	G	A	C	**A**	A	A	T	**T**	G	**C**	T	T	A	G	C	**G**	G	A	A	A	**A**	T	G	G	A	G
**Amino Acid Position**	1 2 8	1 5 9	1 8 8	1 8 9	**1** **9** **1**	2 5 6	2 6 1	2 6 4	**3** **3** **0**	4 3 5	**5** **0** **6**	5 0 8	5 3 8	5 7 4	6 3 1	6 5 5	**6** **8** **0**	6 8 9	7 1 9	7 5 7	7 7 5	**7** **7** **6**	8 1 7	8 3 1	8 9 3	9 3 4	9 4 5
**Common**	C	K	G	E	**G**	C	N	L	**I**	D	**V**	K	D	R	E	L	**M**	L	V	D	S	**E**	V	T	Q	R	K
**Variation**	S	E	S	A	**S**	Y	S	F	**L**	G	**L**	N	V	K	G	S	**V**	V	I	N	R	**K**	F	A	R	H	E
**Malawi and Mozambique, *n* = 115**	1	-	-	-	1	1	-	-	1	1	1	1	-	1	1	1	12	-	1	1	-	2	1	1	1	-	-
**eBL case, *n* = 63**	1	-	-	-	1	-	-	-	-	1	1	-	-	-	-	1	6	-			-	1		1	1	-	-
**Control, *n* = 52**	-	-	-	-	-	1	-	-	1	-	-	1	-	1	1	-	6	-	1	1	-	1	1	-	-	-	-
**Tanzania, *n* = 55**	-	1	1	-	2	-	1	-	1	-	-	-	2	-	-	-	3	-	-	-	1	-	-	-	-	1	1
**Ghana, *n* = 33**	-	1	1	1	1	-	-	1	1	-	-	-	-	-	-	-	4	1	-	-	-	-	-	-	-	1	1
(B) Synonymous change.																											
**Nucleotide Position**	5 6 4		7 7 7		7 9 2		9 0 3		1 2 2 1		1 5 9 0		1 6 0 2		1 7 4 9		1 9 2 3		2 1 0 6		**2** **3** **4** **6**		2 8 8 2		2 6 3 1		
**Common**	T		A		T		A		T		T		A		T		T		A		**T**		T		A		
**Variation**	G		G		C		G		C		C		G		G		C		G		**C**		C		G		
**Amino Acid Position**	1 8 8		2 5 9		2 6 4		3 0 1		4 0 7		5 3 0		5 3 4		5 8 3		6 4 1		7 0 2		**7** **8** **2**		7 9 4		8 7 7		
**Amino Acid**	G		G		L		E		S		D		P		S		S		G		**Y**		G		L		
**Malawi and Mozambique, *n* = 115**	2		1		-		1		1		1		1		1		1		1		2		2		1		
**eBL Case, *n* = 63**	2		1		-		1		1		1		-		1		-		-		1		1		-		
**Control, *n* = 52**	-		-		-		-		-		-		1		-		1		1		1		1		1		
**Tanzania, *n* = 55**	-		-		-		-		-		-		-		-		-		-		1		-		-		
**Ghana, *n* = 33**	-		-		1		-		-		-		-		-		-		-		-		-		-		

**Table 5 cancers-13-01692-t005:** Genetic differentiation (*Fst*) of the *P. falciparum sera5*.

*P. falciparum sera5*Region/Country	Subject Type	Malawi and Mozambique			Tanzania	Ghana
		All	eBL Case	Control	All	All
(A) Octamer repeat region						
Malawi and Mozambique	Combined	-	0.9910	0.9910	0.0721	**0.0360**
	eBL case	−0.0094	-	0.4595	0.1171	**0.0090**
	Control	−0.0097	−0.0030	-	0.0811	**0.0270**
Tanzania	All subjects	0.0113	0.0085	0.0130	-	0.0811
Ghana	All subjects	0.0291	0.0271	0.0286	0.0104	-
(B) Serine repeat region						
Malawi and Mozambique	Combined	-	0.9910	0.9910	0.0811	**0.0000**
	eBL case	−0.0093	-	0.6216	0.0541	**0.0000**
	Control	−0.0096	−0.0027	-	0.3604	**0.0451**
Tanzania	All subjects	0.0069	0.0100	0.0013	-	0.2072
Ghana	All subjects	0.0202	0.0204	0.0181	0.0039	-
(C) Non-repeat 2562 bp sequence						
Malawi and Mozambique	Combined	-	0.9910	0.9910	0.2973	0.5676
	eBL case	−0.0099	-	0.9279	0.4234	0.5946
	Control	−0.0105	−0.0056	-	0.2342	0.8018
Tanzania	All subjects	0.0011	−0.0002	0.0023	-	0.5586
Ghana	All subjects	−0.0033	−0.0037	−0.0067	−0.0044	-

*Fst* values are shown in the bottom left and *p-*values are shown in the upper right. *p* < 0.05 are shown in bold. eBL: endemic Burkitt Lymphoma.

## Data Availability

The *Pfsera5* sequence data has been uploaded to DDBJ (Accession numbers: LC606291-LC606405). The remaining clinical data and the code used for these analyses are available upon reasonable request from the corresponding authors.

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
