# Peer review of "Assessment of Mixed Plasmodium falciparum sera5 Infection in Endemic Burkitt Lymphoma: A Case-Control Study in Malawi"

_cancers, 2021, doi:10.3390/cancers13071692_

Round 1
Reviewer 1 Report
The study examine and interesting and often neglected potential link between Plasmodium falciparum infection and Endemic Burkitt lymphoma. The study is well designed, executed, and analyzed and the presentation is clear. The present findings are important as it supports and extend two other studies and thereby strengthen the hypothesis of a link between Plasmodium falciparum infection and Endemic Burkitt lymphoma. The weakness of the study is that is confirmatory in its scope and that it does not provide convincing new clues to the mechanisms involved in the interplay between the two diseases. In conclusion: A well performed study which adds to the notion that Plasmodium falciparum infection confers susceptibility to Endemic Burkitt lymphoma or enhanced disease progression - an interesting linkage which is not only of interest to researchers in the field of Burkitt lymphoma but more general to researchers in the field of hematologic cancers.
Author Response
Response: We thank the reviewer for their positive comments about our work. We agree with the reviewer that this work is confirmatory to our other two studies. This confirmation was needed to increase our confidence in the notion that genetic complexity of parasites constitutes an exposure that can be studied in relation to eBL risk.
We have revised the manuscript to make two points that were not quite obvious. First, as noted above in our response to the limitations, we stress that our results suggest that PfSERA5 might a good marker to study Pf diversity. i.e.,
“The general finding that PfSERA5 does not show antigenic variation [20] suggests that PfSERA5 might be a good tool to characterize genetic complexity of parasite population in eBL cases and controls. “
Second, we also note that SEAR5 might explain the patterns we observed by contributing to immune tolerance on as described on page 13
“we speculate that PfSERA5 may contribute to molecular camouflage that facilitates tolerance of low-grade infection and sets the stage for progression to cancer. PfSERA5 N-terminal domain decorates the surface of merozoite where it tightly binds to host serum protein, vitronectin, which in turn binds to other host protein that camouflages the para-site from the host immune system [47], potentially reducing pressure for mutations in PfSERA5 gene.”
Reviewer 2 Report
The authors have conducted a case-control study to examine the influence of the complexity of malaria infection on the development of Burkitts Lymphoma, specifically the role that the PfSERA5 gene has on the outcome. I have a few questions and concerns that I lay out below:
- It is unclear at what time point the authors take a blood sample from the participants to assess malaria infection. Can the authors please clarify?
- I’m confused by and concerned with the comparison to Ghana and Tanzania. I don’t quite understand the purpose for these comparisons. Can the authors please specify their objective in doing so? I’m also concerned about the validity of these comparisons with the cases and controls from Malawi and Mozambique. The cases and controls of Mozambique likely represent a much broader geography, being drawn from throughout the region to the cancer center. From what I can tell the Ghana and Tanzania samples are from specific communities and would be more reflective a single location than the Malawi + Mozambique samples. I would recommend dropping these analyses from the manuscript due to their inherent biases. Further I am not sure they add much to the paper that the authors have not already published, and so would recommend dropping this analysis.
- The authors report pooling the samples from Mozambique. On first read this was a bit confusing, did the authors pool the blood from the children from Mozambique or simply analyze the Mozambique samples alongside the Malawi samples without regard to country of origin. Can the authors please clarify?
Author Response
- It is unclear at what time point the authors take a blood sample from the participants to assess malaria infection. Can the authors please clarify?
Response: We have revised the manuscript on page 3, 2nd last paragraph as follows “The current study included only children who were Pf-infected based on polymerase chain reaction (PCR) performed on samples taken at the time of diagnosis before initiating cancer treatment targeting a 519 base pair (bp) segment of PF07-0076 locus” and on Page 4 in the first paragraph “…coded/masked whole blood samples taken at the time of diagnosis before initiating cancer treatment using…”
- I’m confused by and concerned with the comparison to Ghana and Tanzania. I don’t quite understand the purpose for these comparisons. Can the authors please specify their objective in doing so? I’m also concerned about the validity of these comparisons with the cases and controls from Malawi and Mozambique. The cases and controls of Mozambique likely represent a much broader geography, being drawn from throughout the region to the cancer center. From what I can tell the Ghana and Tanzania samples are from specific communities and would be more reflective a single location than the Malawi + Mozambique samples. I would recommend dropping these analyses from the manuscript due to their inherent biases. Further I am not sure they add much to the paper that the authors have not already published, and so would recommend dropping this analysis.
Response: We agree with the reviewer’s observations and concerns. However, given the paucity of molecular data about Pf conducted in the context eBL studies, we believe there is value in reporting these results that were generated in the same lab following the same method. However, while we have not agreed with the reviewer’s suggestion to drop this entire comparison, we have taken the reviewer’s suggestion to be more open about the limitations of the data. In addition to our original transparent description of the source of samples, we have revised the 4th paragraph of the discussion on page 14.
“We also observed that PfSERA5 diversity was not unique to Malawi + Mozambique samples or to eBL cases. These results are reassuring because our comparisons are based on samples obtained using different methods, which may be biased. The results from Ghana and Tanzania are from specific communities living in relatively small geographical areas, whereas the Malawi + Mozambique data represent a sampling of parasites from a much larger geographical area. Thus, the observation that no specific Pf variants or haplotypes are associated with eBL and that both eBL cases and controls in Malawi harbour common haplotypes is consistent with both groups being exposed to similarly high environmental pressure of Pf infection [35]. However, we note that rare haplotypes may be harder to detect because they exist at lower submicroscopic levels. Our findings of a lower frequency of the common haplotypes in Tanzania and Ghana than in Malawi + Mozambique may be due to geographic confounding as noted above.”
- The authors report pooling the samples from Mozambique. On first read this was a bit confusing, did the authors pool the blood from the children from Mozambique or simply analyze the Mozambique samples alongside the Malawi samples without regard to country of origin. Can the authors please clarify?
Response: We agree with the reviewer that our sentence is ambiguous. We have revised as follows on Page 5
“Because of the small sample size of the children originating in Mozambique, the sequence data from those children were analyzed together with those from Malawi, where all children were enrolled.“
Reviewer 3 Report
This is an excellent manuscript, well constructed and well described despite quite technical methods. I think the authors did a really great job of putting the results in context of the literature with good tables and figures describing the results.
The only minor revision I request is a bit of information on the Pf SERA5 locus and why it is only found in about half of patients with positive Pf PCR. I assumed it was a universal protein found in all malaria genomes based on the way the intro was written. I may have missed it but perhaps clarifying why it is only found in about half of hte participants would be very helpful.
Great work!
Author Response
- The only minor revision I request is a bit of information on the Pf SERA5 locus and why it is only found in about half of patients with positive Pf PCR. I assumed it was a universal protein found in all malaria genomes based on the way the intro was written. I may have missed it but perhaps clarifying why it is only found in about half of hte participants would be very helpful.
Response: The reviewer is correct that although we studied only children confirmed to be Pf infected, only half were found to be positive using PfSERA5 PCR. We suspect that this is because PfSERA5 is a much larger fragment (3.2 Kb versus the 519 bp fragment) than that used in the original study). Amplifying 3.2 kb (Pf-SERA5) requires higher quality higher volume gDNA than amplifying 519 bp (Pf positive).
We have revised manuscript as follows on Page 7
“Likewise, not all Pf positive samples was suitable for the amplification of the near full-length PfSERA5 due mainly to the quality of samples available for the study.”
In addition, we have revised the section about limitations as follows on Page 14
“The general finding that PfSERA5 does not show antigenic variation [20] suggests that PfSERA5 might be a good tool to characterize genetic complexity of parasite population in eBL cases and controls. However, this should be balanced against the likelihood that sample size will shrink because amplifying about 3.2 Kb of PfSERA5 requires higher quality gDNA than amplifying 519 bp that was used to confirm Pf infection in all these samples. This explains why our sample size shrunk by 50%.”